# Predicting regrowth of low-grade gliomas after radiotherapy

**Stéphane Plaszczynski**[1,2]*, **Basile Grammaticos**[1,2], **Johan Pallud**[3,4,5], **Jean-Eric Campagne**[1,2], **Mathilde Badoual**[1,2]

**1** Université Paris-Saclay, CNRS/IN2P3, IJCLab, Orsay, France, **2** Université Paris-Cité, IJCLab, Orsay, France, **3** Department of Neurosurgery, GHU Paris Sainte-Anne Hospital, Paris, France, **4** Université de Paris, Sorbonne Paris Cité, Paris, France, **5** Inserm, U1266, IMA-Brain, Institut de Psychiatrie et Neurosciences de Paris, Paris, France

* stephane.plaszczynski@ijclab.in2p3.fr

## Abstract

Diffuse low grade gliomas are invasive and incurable brain tumors that inevitably transform into higher grade ones. A classical treatment to delay this transition is radiotherapy (RT). Following RT, the tumor gradually shrinks during a period of typically 6 months to 4 years before regrowing. To improve the patient's health-related quality of life and help clinicians build personalized follow-ups, one would benefit from predictions of the time during which the tumor is expected to decrease. The challenge is to provide a reliable estimate of this regrowth time shortly after RT (i.e. with few data), although patients react differently to the treatment. To this end, we analyze the tumor size dynamics from a batch of 20 high-quality longitudinal data, and propose a simple and robust analytical model, with just 4 parameters. From the study of their correlations, we build a statistical constraint that helps determine the regrowth time even for patients for which we have only a few measurements of the tumor size. We validate the procedure on the data and predict the regrowth time at the moment of the first MRI after RT, with precision of, typically, 6 months. Using virtual patients, we study whether some forecast is still possible just three months after RT. We obtain some reliable estimates of the regrowth time in 75% of the cases, in particular for all "fast-responders". The remaining 25% represent cases where the actual regrowth time is large and can be safely estimated with another measurement a year later. These results show the feasibility of making personalized predictions of the tumor regrowth time shortly after RT.

## Author summary

This work addresses the question of making predictions on the remission time patients suffering from brain tumors (gliomas) may expect after a radiotherapy treatment. It is a very crucial question often asked by patient's to their practician in order to plan some important life projects (as traveling, retiring, having children) but that is difficult to answer since there is a large variability among patient reactions to radiotherapy. We then build a statistical model using the recorded evolution of the glioma size of previously

**Data Availability Statement:** All data necessary to reproduce the work (Fig 2, Tables 1, 2, Eqs 4, 6) are available in the manuscript. python3 code to reproduce Figs 6, 7, and 8 is provided in Supporting information.

**Funding:** The author(s) received no specific funding for this work.

**Competing interests:** The authors have declared that no competing interests exist.

treated patients. It allows, for the first time, to make these predictions, with a few months accuracy.

This is a *PLOS Computational Biology* Methods paper.

## 1 Introduction

Diffuse gliomas are primary brain tumors originating from glial cells (oligodendrocytes and/or astrocytomas). In its 2016 classification, The World Health Organization defines four grades [1]: while the first grade gliomas are benign, second grade gliomas (or low grade gliomas, LGG) are invasive, growing at a rate of 2 to 8 mm/year [2] in diameter, but without involving metastasis or necrosis. Unfortunately, they cannot be cured by oncological treatments [3] so one needs to contain their growth as long as possible, before they transform into grade III and IV (glioblastomas) with a dramatically low survival rate. LGG are detected with magnetic resonance imaging (MRI) scans under a T2-FLAIR sequence. Since they are diffuse tumors that extend beyond the observed boundaries [4, 5], the uncertainty on their size is irreducible. Classical treatments include resection (when possible), chemotherapy and radiotherapy (RT) [6].

Standard conformational radiotherapy for LGG is generally performed during 6 weeks (5 days a week) and the classical dose is around 50 Gy. Irradiation of gliomas involves a large number of physical processes [7] and its effect varies across patients. However, some general features emerge: the tumor shrinks during a period that varies between a few months and several years, before regrowing at a rate similar to the one observed before radiotherapy.

Mathematical modeling of natural and under treatment tumor growth has a long and rich history (in particular for gliomas, one can refer to the recent review [8]). For invasive tumor such as gliomas that cannot be removed by surgery, one aspect that is of special interest for clinicians is the response of tumor to treatments and in particular, radiotherapy [9–13]. Its primary goal is to optimize treatments "virtually": for example, choosing the optimal radiation fraction of doses [14], finding the best way to combine it to chemotherapy [15] or studying its interplay with the immune system [16]. Beyond describing qualitatively the different processes at stake, the real usefulness of a model would be to predict the response of individual patients to a treatment, even before the end of the treatment. Such predictions would allow the clinician to personalize the follow-up (and the treatment) for each patient. There has been some attempts to predict tumor growth and the effects of treatments on individual patients. If purely statistical or image-based models can be used to predict glioma growth [17], mechanistic models are usually used for instance to predict the metastatic relapse in breast cancer [18], tumor growth in leukaemia and ovarian cancer [19], response of high grade gliomas to chemoradiation [20], or the patient-specific evolution of resistance in the context of prostate cancer [21].

For low-grade gliomas, individualized predictions from the tumor size dynamics and genetic characteristics, have been made for the response to a chemotherapy treatment [22]. To our knowledge, such individual predictions do not exist in the case of low-grade glioma and RT. In this article, we show that it is possible to predict the evolution of LGGs under RT, for individual patients, with an approach based on a practical mechanistic model, even in the case where the number of patients is not sufficient to apply standard machine-learning techniques.

In order to be used for predictions, a model should have a limited number of parameters. Models that are too detailed are useful to describe qualitatively the tumor evolution but usually involve too many unknown parameters [23]. Given the scarcity of clinical data, only a small number of parameters can be deduced. The goal here is to keep as few as possible parameters but still to capture the essential dynamics of the tumor.

In a previous work [13], we analyzed a large number (43) of LGG radial evolutions under RT and proposed a physically motivated model, with 4 parameters, that fitted well all the profiles of tumor evolution during patient's follow-up. Following that work, we now try to make predictions using that model. This is challenging given the variety of possible in-vivo reactions to radiation. We have chosen to focus on the moment when the tumor stops shrinking and starts to regrow, what we call in the following the "regrowth time". This is an essential feature of the tumor dynamics for two reasons. First, the patients often ask their clinician when the tumor will regrow in order to plan some major life projects (as having a child, traveling, retiring, etc.). This would be a valuable information to improve their life-quality. Second, the the dates for the next MRIs are currently fixed and not optimal on an individual basis. By making predictions we may adjust them more precisely for personalized follow-ups.

## 2 Materials and methods

### 2.1 Ethics statement

The study received required authorizations (IRB#1: 2021/20) from the human research institutional review board (IRB00011687). The requirement to obtain informed consent was waived according to French legislation (observational retrospective study).

### 2.2 The patients

We had at our disposal a set of 43 patients with LGGs who were diagnosed at the Sainte-Anne Hospital (Paris, France) from 1989 to 2000. These patients were selected according to precise criteria that are detailed elsewhere [24]. In short, only adults with typical LGGs (that is, no angiogenesis and, thus, no contrast enhancement on gadolinium-T1 images), available clinical and imaging follow-ups before, during, and after RT, and RT as their first oncological treatment except for stereotactic biopsies were eligible. The external conformational RT was given using the same methodology (total dose, 50.4–54 Gy; 6-week period) at 2 outside institutions. The patients had an MRI follow-up before, during, and after RT. Three tumor diameters in the axial, coronal, and sagittal planes on each MRI image with T2-weighted and FLAIR sequences were measured manually. The mean radiological tumor radius was defined as half the geometric mean of these three diameters and was measured as a function of time. The error bars for the measured mean radii were estimated by clinicians and were set to ±1 mm. From this cohort, we discarded the patients that did not have any sign of tumor regrowth at the last time point or those that had fewer than five time points in their follow-up.

### 2.3 The model

A biologically motivated model with the effect of RT on LGG has been presented in [13] and validated by the fits on 43 patient follow-ups. It is based on a standard diffusion-proliferation equation [25] and RT is modeled with a time-dependent death rate ($\kappa_D(t)$). The evolution of the glioma cell density then follows the equation

$$\frac{\partial \rho}{\partial t} = D\Delta\rho + [\kappa - \kappa_D(t)]\rho(1-\rho), \tag{1}$$

where $\rho(r, t)$ is a function of the radius $r$ (assuming a spherical symmetry) and time $t$ (conventionally set to zero at the beginning of RT), $D$ is the diffusion coefficient and $\kappa$ the proliferation rate. In its most simple (thus predictive) form, the death-term is characterized by an amplitude and a characteristic time

$$\kappa_D(t) = \kappa_d e^{-t/\tau_d} \quad \text{for } t \geq 0, \tag{2}$$

and is considered as null before RT.

Assuming that the tumor growth-rate when patients consult is already in the asymptotic state, i.e. that it evolves linearly with a speed $v = \sqrt{2D\kappa}$, and neglecting diffusion after RT, the radius evolution can be approximated by [13]

$$R(t) = R_0 + vt - v\tau_d \frac{\kappa_d}{\kappa}(1 - e^{-t/\tau_d}). \tag{3}$$

Inspired by this formula, we simplify the model Eq (1) by proposing a purely geometrical one in the form

$$R(t) = R_0 + vt - k(1 - e^{-t/\tau}). \tag{4}$$

which has 4 free parameters: $R_0$, $v$, $k$, $\tau$. They cannot be related to the ones obtained by solving numerically Eq (1) since Eq (3) neglects diffusion. This simple geometrical model has the considerable advantage of being analytical. The role of each terms is clear and sketched in Fig 1. It captures the 3 phases of the evolution: first the linear growth, then the exponential decay of a fraction of the tumor and therefore of its radius, third, the regrowth with the same velocity as before RT. This phenomenological model is similar to one proposed for the evolution of prostate cancer [26], although in our case the tumor grows linearly with time before RT, while the prostate one grows exponentially.

To test whether this model does fit our data appropriately, we construct a classical objective function as the mean squared error from the set of measured values $\{t_i, R_i\}$

$$\chi^2(R_0, v, k, \tau) = \sum_{i=1}^{N}[R_i - R(t_i; R_0, v, k, \tau)]^2/\sigma_i^2 \tag{5}$$

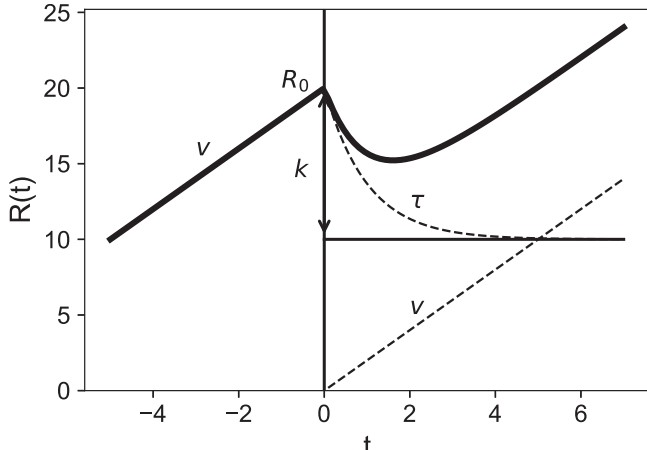

**Fig 1. Illustration of the analytical model describing the tumor radial evolution.** Before RT ($t < 0$), the radius evolves linearly with the asymptotic speed $v$ and reaches $R_0$ at $t = 0$. Then it becomes the sum of an exponential decay of amplitude $R_0 − k$ and characteristic time $\tau$ (death term) and a linear $vt$ regrowth.

**Table 1. Parameters of the least-square solutions of Eq (5) corresponding to the fits shown on Fig 2.** The first columns represents the patients' ID, $(R_0, v, k, \tau)$ are the estimated parameters of the model, and the regrowth time $(t_{min})$ is derived from them. Lengths $(R_0, k)$ are expressed in mm and times $(\tau, t_{min})$ in years.

| id | $R_0$ | $v$ | $k$ | $\tau$ | $t_{min}$ |
|----|-------|-----|-----|--------|-----------|
| (0) | 17.95 | 0.50 | 7.29 | 1.98 | 3.95 |
| (1) | 29.03 | 1.33 | 10.49 | 0.19 | 0.72 |
| (2) | 24.56 | 3.24 | 17.69 | 1.40 | 1.91 |
| (3) | 16.86 | 0.96 | 13.28 | 1.57 | 3.42 |
| (4) | 25.40 | 1.11 | 5.20 | 0.10 | 0.39 |
| (5) | 28.72 | 1.37 | 10.78 | 0.47 | 1.33 |
| (6) | 27.00 | 1.99 | 17.00 | 2.39 | 3.04 |
| (7) | 23.26 | 1.23 | 5.92 | 1.18 | 1.66 |
| (8) | 15.83 | 2.45 | 6.84 | 0.45 | 0.82 |
| (9) | 31.60 | 1.13 | 8.35 | 0.38 | 1.12 |
| (10) | 26.45 | 4.00 | 16.15 | 0.45 | 0.98 |
| (11) | 14.67 | 1.54 | 4.66 | 0.32 | 0.72 |
| (12) | 41.20 | 4.00 | 30.74 | 1.83 | 2.63 |
| (13) | 16.64 | 3.59 | 13.83 | 1.06 | 1.37 |
| (14) | 20.21 | 1.30 | 14.84 | 3.64 | 4.16 |
| (15) | 19.33 | 0.70 | 7.80 | 0.81 | 2.11 |
| (16) | 23.61 | 3.37 | 23.15 | 2.28 | 2.51 |
| (17) | 32.68 | 0.72 | 7.30 | 0.43 | 1.36 |
| (18) | 35.02 | 2.19 | 15.32 | 0.99 | 1.93 |
| (19) | 28.06 | 0.52 | 10.16 | 1.29 | 3.52 |

where $\sigma_i = 1$ mm. This is a 4-parameter real valued function that we minimize easily with a standard optimization algorithm [27] since the model is analytical. To obtain physical results, we impose as limits that all parameters be positive and that the radial asymptotic speed lie in the range $0.5 \leq v \leq 4$ mm/yr [2]. We then obtain the best-fit parameter values given in Table 1 and show, in Fig 2, the comparison between the data and the fitted model on a set of 20 patients who possess at least 9 data points. The agreement is excellent. Although the results are quite similar to the ones obtained in [13], we have considerably simplified the model and reduced drastically the run-time which will be useful later in making predictions.

## 2.4 Constraining the parameters space

We now study whether some common features appear in our best-fit parameters. Fig 3 shows the histograms for each parameter on the 20 patients. No parameter displays a clearly peaked distribution. For a given patient, the expected parameters are random variables and a priori unpredictable, although within some bounds. We then consider the correlation between the variables by computing their Pearson coefficients and show the results in Table 2. The structure is far from being diagonal, indicating non-trivial correlations among most pairs of variables. Of particular interest is the large $(k, v)$ correlation since it relates a quantity defined before RT $(v)$ to a one after RT $(k)$.

To make use of the information in the most efficient way, we first decorrelate the variables. This is performed by diagonalizing the covariance matrix, which is always possible since the covariance matrix is by construction always positive-definite. From the eigenvectors, we build the transformation matrix $T$ that projects our parameters $\boldsymbol{p}^T = (R_0, v, k, \tau)$ onto an orthogonal

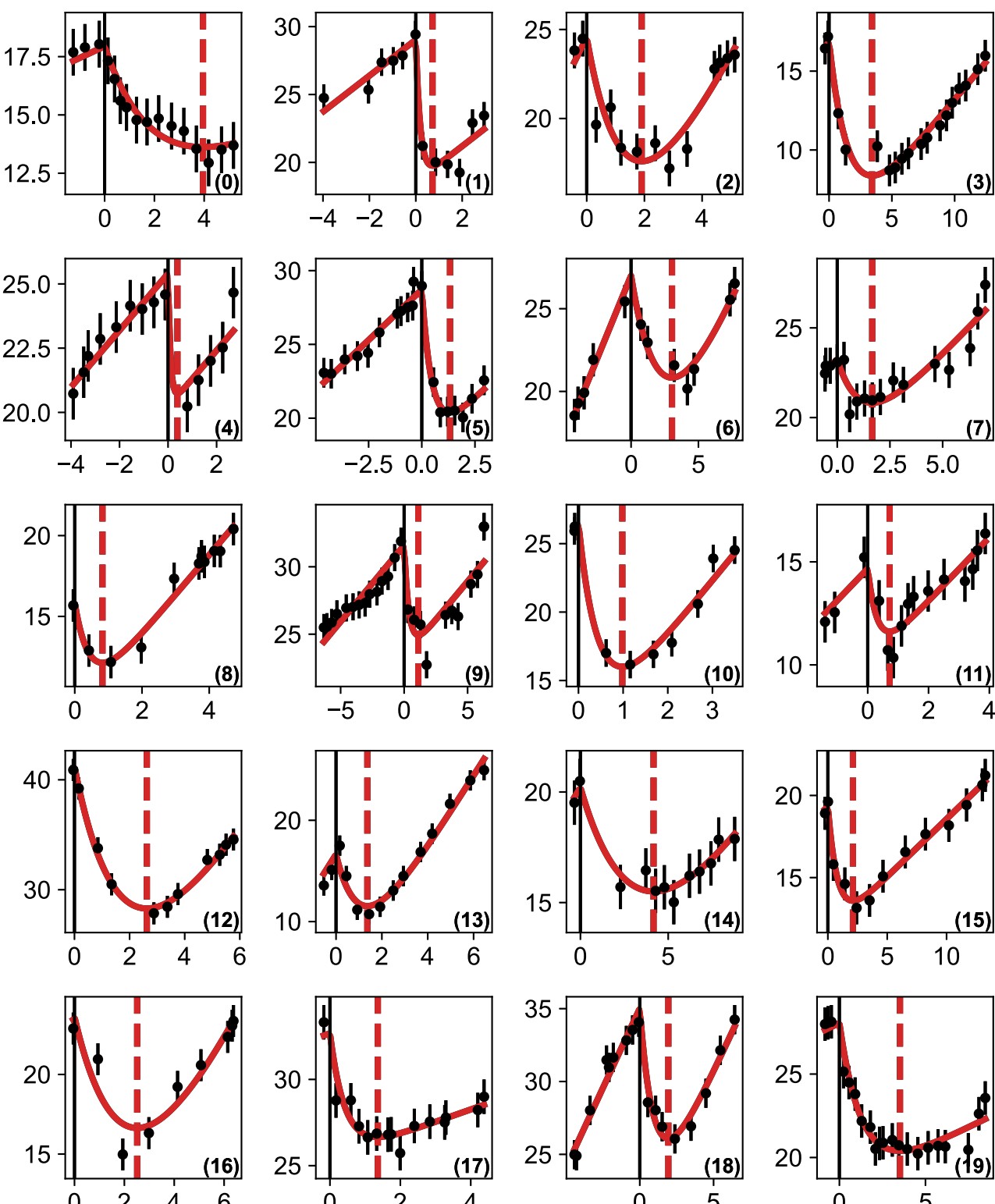

**Fig 2. Comparison between the measured values of the tumor radius and the bestfit model for 20 patients.** The points represent the measured values and the red line our model obtained by minimizing Eq (5). The abscissa represent time in years (with the origin set at RT) and the ordinate the tumor radius (in mm). The error bars on the measurements are of 1 mm. The dashed vertical red line shows the model minimum, i.e. the moment regrowth starts.

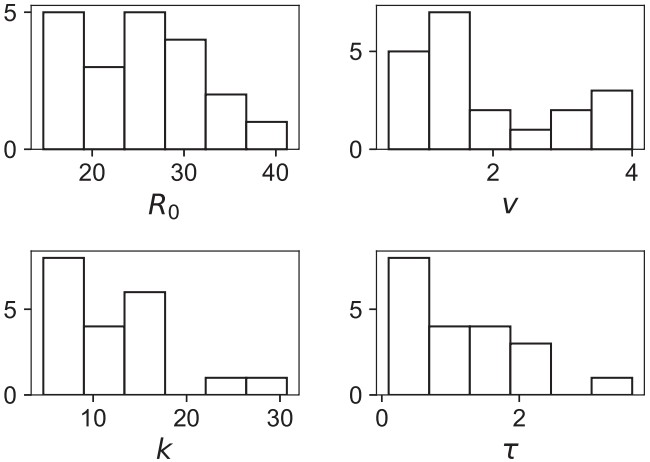

**Fig 3. Histograms of the bestfit parameters.**

basis where the new variables $X^T = (x_1, x_2, x_3, x_4)$ are uncorrelated. From our data we measure the following projection matrix:

$$T = \begin{pmatrix} 0.75 & 0.07 & 0.65 & 0.02 \\ 0.65 & -0.13 & -0.74 & -0.11 \\ -0.06 & -0.64 & 0.17 & -0.74 \\ 0.01 & -0.75 & 0.05 & 0.66 \end{pmatrix} \tag{6}$$

and the linear change of variables is then simply

$$X = Tp \tag{7}$$

Considering the important terms in the matrix, we see that the first 2 lines link essentially the size of the tumor ($R_0$) to the amplitude of the RT reaction ($k$). The next two ones relate in a non-trivial way, the growth speed ($v$) to the RT effect ($k$, $\tau$).

We now consider the distribution of these new $\{x_{i=1,\dots,4}\}$ variables that which, we recall, are mutually uncorrelated by construction. Their histograms are shown on Fig 4.

The nice feature now is that, unlike the original variables (Fig 3), the distributions are now approximately Gaussian Although it is a slightly questionable assertion for $x_2$, the standard deviation of the fit is large enough to capture reasonably all the points. For each variable we fit the mean ($\mu_i$) and standard-deviation $\sigma_i$.

**Table 2. Correlation coefficients measured between the 20 bestfit parameters of our model.** Since the matrix is symmetric with ones on the diagonal we only show its upper half.

|       | $R_0$ | $v$  | $k$  | $\tau$ |
|-------|-------|------|------|--------|
| $R_0$ | 1     | 0.17 | 0.46 | -0.09  |
| $v$   |       | 1    | 0.73 | 0.10   |
| $k$   |       |      | 1    | 0.52   |
| $\tau$|       |      |      | 1      |

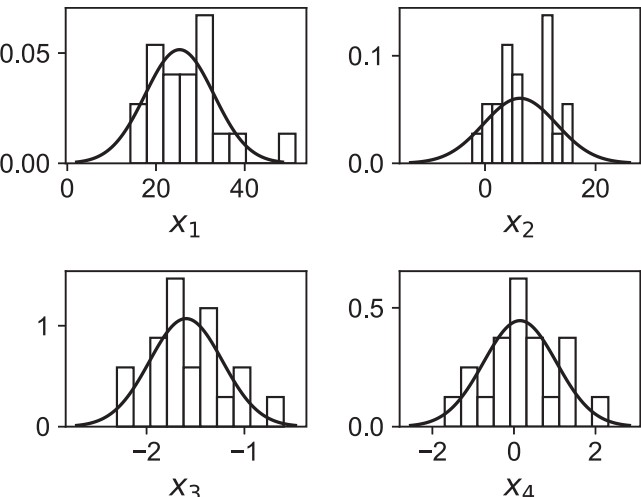

**Fig 4. Histograms of the transformed decorrelated variables.** The new variables are linear combinations of the fitted $(R_0, v, k, \tau)$ parameters as described in the text. They are normalized to unit area and the result of the Gaussian fit is shown in black.

We can now build a term that contains the extra-information about the correlations among the variables in the form

$$\chi^2_{cons}(R_0, v, k, \tau) = \sum_{i=1}^{4} \left[ \frac{(x_i(R_0, v, k, \tau) - \mu_i)}{\sigma_i} \right]^2,$$  (8)

where the $x_i$'s are computed according to Eq (7), and $(\mu_i, \sigma_i)$ are the parameters of the Gaussian fits shown on Fig 4.

We then add this term to the original $\chi^2$ function (Eq (5))

$$\chi^2_{TOT}(R_0, v, k, \tau) = \chi^2(R_0, v, k, \tau) + \chi^2_{cons}(R_0, v, k, \tau)$$  (9)

and perform the minimization. The constraint acts as a Bayesian prior, i.e. it includes all the a priori information we have between the parameters. It should not be used it on the previous Fig 2 fits (since it was derived from them), but we checked that the best-fits obtained using $\chi^2_{TOT}$ are *exactly* the same as the ones with only the $\chi^2$ term, meaning that we are not over-constraining the parameters with the constraint.

So why add such a term? Suppose we have few data, for instance 2 measurements before RT and one after, then we have only 3 points to determine 4 parameters. Using Eq (8) we introduce some extra equations and the problem becomes at least technically solvable.

In the following we focus on the regrowth time, which according to our model (Eq (4)) is

$$t_{min} = \tau \ln \left( \frac{k/\tau}{v} \right).$$  (10)

It depends mostly on $\tau$ and logarithmically on the relative speed between the shrinkage due to RT ($k/\tau = v_d$) and the intrinsic tumor growth ($v$). The $\chi^2_{TOT}(R_0, v, k, \tau)$ minimization leads to the $(\hat{R}_0, \hat{v}, \hat{k}, \hat{\tau})$ estimates and we use those values in Eq (10) to estimate the regrowth time.

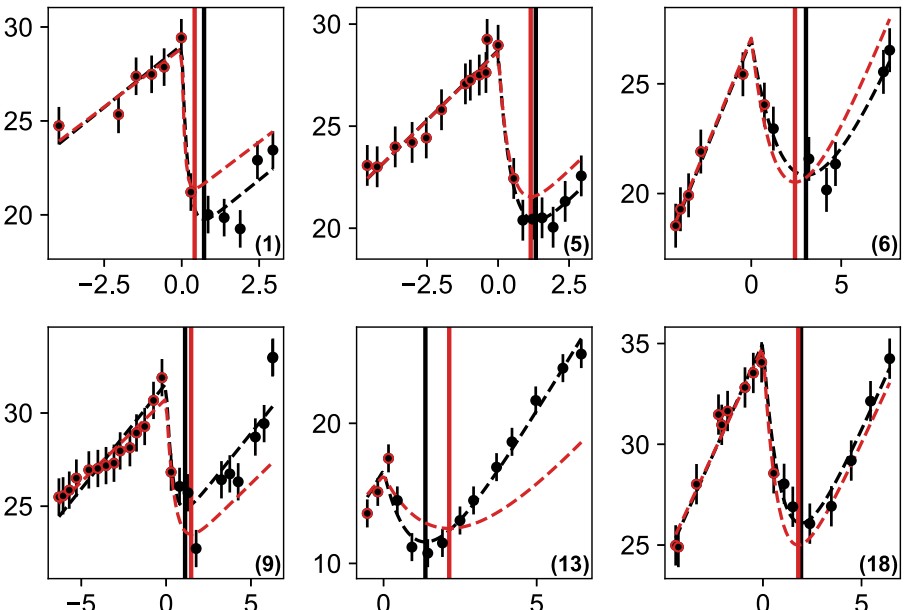

**Fig 5. Predictions for the regrowth time on real data with the first point after RT.** The black dashed curve shows the best fit model using all the data points and the vertical black line shows its minimum (same as Fig 2). We then only consider the red circled points consisting of all the points before RT and the first one just after, and perform the constrained minimization described in the text. The result for the model is the dashed red curve with the estimated regrowth time shown as the vertical red line.

## 3 Results

### 3.1 Data validation

We first validate the procedure on our dataset by assessing the performances of our predictions with a *single* point after RT.

Among our patients, we choose 6 follow-ups, with at least two points before RT and enough subsequent points for the minimum of the fit to be robust (see Fig 2). We then take the points before RT and the first one just after it, and perform the constrained minimization (Eq (9)). We obtain an estimate of $t_{min}$ and compare it to the one from the full fit. In order to avoid mixing the training and test samples, for each patient we rebuild the constraint on the lines of section 2.4, removing each time the patient's data from the datasets. The results are shown in Fig 5.

The $t_{min}$ predictions for most of the patients lie within a few months of the value determined with all data, which is quite successful given the little amount of information and the fact that each patient react differently to the treatment. For patient (13) it is slightly larger (10 months). This is an interesting case, since the point after RT is *above* the one before. This can be due to statistical fluctuations or to the fact that RT produces sometimes an oedema that can be misidentified as the tumor radius. However even in this case, we obtain a reasonable estimate. This shows that, at the date of the first MRI after RT, we could have guessed in most cases efficiently the regrowth time of the tumor and plan more efficiently the dates of the next MRIs.

### 3.2 Predictions

We now evaluate on virtual patients a strategy to estimate as soon as possible the tumor regrowth time. To this aim, we must first fix the times of the MRI measurements which are constrained in the following way.

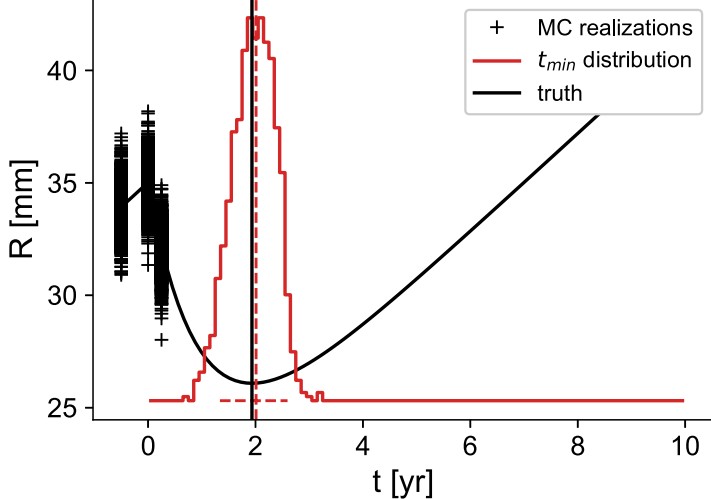

**Fig 6. Characterization of there growth time estimation with a Monte-Carlo method.** The black curve represents a model (which is here the bestfit of patient (18)) with its minimum shown as the vertical black line ("truth"). One draws some Gaussian noise of $\sigma = 1$ mm at the measurement times $t_{mes} = [-6, 0, 3]$ months, and performs the $t_{min}$ estimation described in the text. This is repeated 1000 times which allows to construct the red histogram of all the $t_{min}$ estimates. The red vertical dashed line shows its mean value and the horizontal one the [0.05,0.95] percentile region.

1. Although we showed results with many points before RT (Fig 5), today's clinical paradigm is to reduce the tumor as soon as possible. We thus consider the case where the radiotherapy sessions are planned immediately after the first MRI within typically 6 months.

2. Since this is a central point, a second MRI should be performed around the RT date.

3. Fast-responders reacting within a few months, we propose to perform an MRI measurement 3 months after RT.

We will then consider the cases where the measurement times are located at $t_{mes} = [-6, 0, +3]$ months and test if we can still make some predictions for the regrowth time. This is a very challenging situation since we only have 3 nearby points with important relative errors. To assess statistically the performances of the prediction, we adopt a Monte-Carlo approach. For a given set of "true" parameters $(R_0, v, k, \tau)$, we first compute the tumor radius at $t_{mes}$. We then add to each point a random Gaussian noise with a $\sigma = 1$ mm standard deviation, and from these virtual measurements, estimate the regrowth time. Since we noticed that a few hundreds of iterations is sufficient to reach a stable distribution, we repeat the procedure 1000 times. We then consider the mean of the predictions and the 95% confidence-level interval (obtained from the [0.05, 0.95] percentiles) that we compare to the true $t_{min}$ value. This procedure is illustrated in Fig 6.

We use our 20 best-fits as a representative set of "true models". We perform the Monte-Carlo study described previously for each set of parameters and compare the mean and 95% confidence-level interval of our estimated regrowth times to the true value on Fig 7.

First, we notice that 15 predictions out of 20 (75%) are good, the mean value being typically within 6 months of the true. In these cases, the guess follows roughly the true values which confirms that the method is not only driven by the constraint (which would lead always to the same interval) but also incorporates the information of the 3 measurements. Fast-responders (patients (1),(4),(8) and (11)) are correctly predicted and tend to lead to predictions under 1 year which could be the threshold to plan a next MRI rapidly (possibly 3 months later).

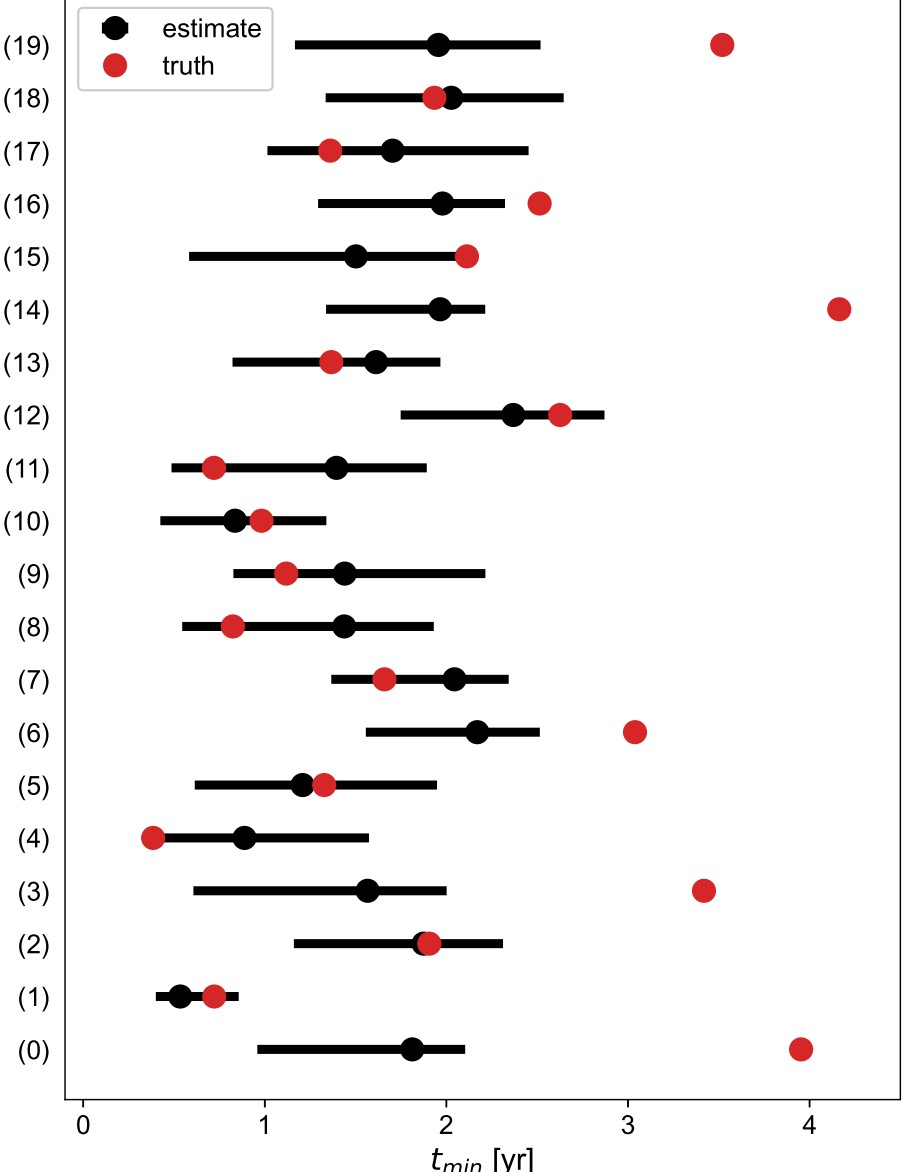

**Fig 7. Performances of the regrowth time estimates with 3 measurements at $t_{mes} = [-6, 0, 3]$ months for a set of true parameters corresponding to the bestfits of our 20 patients.** Black points represent the mean of the estimates and the bars the 95% confidence-level interval. The red point is the true value associated to each best-fit for the patients labeled on the vertical axis (corresponding to the dashed lines in Fig 2). Note that only the best-fit parameters of each patient are being used here.

There are also 5 outliers out of 20 (25%) corresponding to the cases where the true regrowth times are the largest (3–4 years), i.e. to the *slowest* responders. A point at 3 months for them is much too soon to infer *any* information about the curvature, so that the prediction is only driven by the constraint and goes to its mean value of about 2 years. More precisely, by Taylor-expanding our model near $t = 0^+$

$$R(t) = R_0 + (v - v_d)t + \epsilon t^2/2 + \mathcal{O}(t^3) \qquad (11)$$

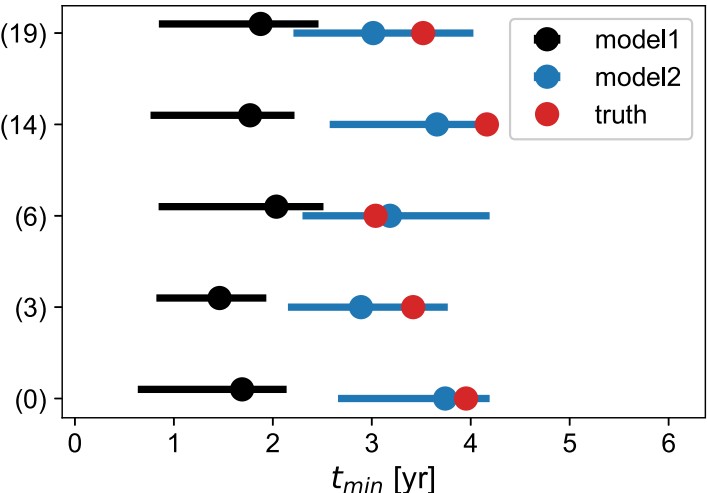

**Fig 8. Monte-Carlo estimates of the regrowth time with an extra point at 1 year for the 5 outliers of Fig 7 ($t_{min} > 3$ years).** The black bars (model 1) corresponds to the 95% confidence-level intervals obtained from the standard constraint (Eq (8)) and the blue ones (model 2) with the loose constraint described in the text. The black/blue points corresponds to the mean values and the red point is the true value of each model.

where $v_d \equiv k/\tau$ is the speed of the collapse and the curvature term is $\epsilon = v_d/\tau$. For slow-responders, there is almost no curvature at 3 months, $\epsilon \to 0$ and $\tau = v_d/\epsilon$ diverges leading to a very broad (and even sometimes bi-modal) $t_{min}$ distribution. In this case, the prediction is only driven by the constraint.

Although pessimistic for the patient, the predicted value is still *large* (around 2 years, see Fig 7). Thus we can safely plan a next MRI 1 year after RT. We consider the case where the times for the radial measurement are at $t_{mes}$ = [−6, 0, 3, 12] months and perform the prediction again. The result is shown in black in Fig 8.

Unfortunately, the constraint Eq (8) is still pulling $t_{min}$ to too low values. We need to switch to a looser constraint. As is clear from Eq (11), the linear term, that is the best constrained, is related to the slopes measured before ($v$) and after ($v_d$) RT. In the absence of good knowledge of the curvature, we may try to relate these slopes to the regrowth time. Indeed, on our dataset, we observe a strong correlation between $v_d$ and $t_{min}$ (Fig 9) that we fit to a power-law

$$t_{min} = 9.0/v_d^{0.61}. \tag{12}$$

This correlation, that emerges from the data, is highly non-trivial. According to Eq (10)

$$t_{min} = \tau \ln\left(\frac{v_d}{v}\right). \tag{13}$$

If $\tau$, $v$ and $v_d$ were uncorrelated, the regrowth time would raise logarithmically with $v_d$. This correlation thus relates what happens before RT ($v$) to what happens after ($v_d$, $\tau$). Understanding its nature would require a full biophysical model of radiation effects that is outside the scope of our phenomenological approach.

We can use this correlation to build a new estimator for $t_{min}$: we fit on the data only the linear terms in Eq (11) in order to get $v_d$ which we transform according to Eq (12). Since this method uses a single correlation, we call it the *loose constraint*. We show the result of applying this procedure on the outliers in blue on Fig 8. The distributions are much better centered on

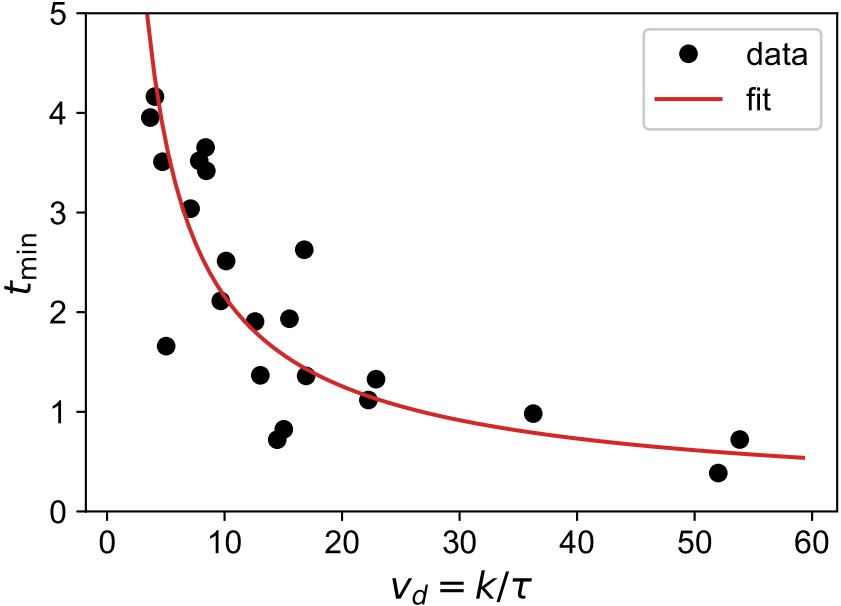

**Fig 9. Correlation between $v_d$ and $t_{min}$ measured on our set of 20 patients.** The red line shows the power-law fit.

the true values. One may ask why not always use this constraint. As clear in the figure, the uncertainty is larger with the loose-constraint method. This is the classical bias/variance trade-off of any estimator. Although it gives indeed less biased results for outliers, the method would miss fast-responders at 3 months, since the slopes determination is then extremely noisy. On the contrary, with a point at 1 year, there is enough lever-arm to determine the slope quite precisely and take advantage from the correlation to let the data "speak for themselves".

We point out that the loose-constraint method is very simple and may be used by any clinician without even a computer. First measure the slope before RT to obtain $v$, then the slope after RT ($v - v_d$) to obtain $v_d$, and finally use Eq (12) to predict the regrowth-time.

## 4 Discussion

We have proposed a new simple model to describe the evolution of diffuse low-grade gliomas before and after radiotherapy. It is analytical and describes in a satisfactory way the follow-ups of 20 patients with measured tumor radii before and after RT. This model has 4 free parameters, 2 before RT and 2 after, that vary for each patient. From the study of the correlation between all the parameters we proposed a way to include a prior information to any follow-up, which allows to perform predictions for the regrowth-time of the tumor rapidly after RT. From the data we had at our disposal, we showed that including this information allows to predict the regrowth time of the tumor at the very first MRI measurement after RT typically within 6 months. Using virtual patients, we have shown that is is possible to predict reasonably well the regrowth time with only one point 6 months before RT, one around RT and one 3 months after, in 75% of the cases. The remaining 25% for which our prediction is *pessimistic*, have all large regrowth-time ($\simeq 4$ years) and may draw benefit from another measurement 1 year after RT, leading to more correct estimates.

These results assume that our database is representative of all LGG evolution and would profit from incorporating more patients' data. Similar profiles are obtained for chemotherapy treatments [22, 28] and it would be interesting to redo the analysis in this case.

This work is based on a 4-parameters model which is a simplified version of a biologically motivated model. This choice can be challenged; why not use some non-parametric method that are often efficient? First, the low dataset (43 patients but in practice 20 with a sufficient number of points to inform our model) precludes the possibility of using general purpose Machine Learning techniques like Deep Neural Networks, Random Forests, Boosted Decision Trees (as described for instance in this recent review [29]), as well as Recurrent Networks dedicated to Time Series (e.g. [30]). Second, we could think of using Gaussian Processes (GP) method (e.g. [31]) that can work on small samples with some optimized kernel. We have tried it, with a squared exponential kernel and a white noise. However, by construction, outside the data input region the naive "vanilla" model converges to a constant and cannot describe the regrowth phase. To overcome this failure, one is forced to use a time dependent function of the mean which is exactly the meaning of the 4-parameters model developed in this article. This clarifies why modeling, especially based on physical arguments, is superior to all purely statistical methods. This was the key to the success of making predictions from a restricted dataset and with very few data points.

Here, we have varied the patient's population and shown that the method has the potential to make some predictions among various patients profile. The problem is different for a personalized follow-up (which is the practical clinical case) since the prediction depends on the details of the measurements (times and values). Using a Monte-Carlo Markov Chain technique, one can obtain an individualized probability distribution of the regrowth-time that can help clinicians adapt their treatment and the dates of the next MRIs.

Nowadays, radiotherapy is no longer used as a first intention main treatment for low-grade gliomas. The actual recommended treatment for high-risk patients is predominantly a combination of radiation and chemotherapy, after surgery whenever feasible [32]. However, an exclusive radiotherapy can still be proposed for high-risk low-grade gliomas when surgery and chemotherapy are not feasible. It would be interesting to study whether our simple model could be adapted to model the effect of the combination on low-grade gliomas (without surgery).

## Supporting information

**S1 Code. python3 software to reproduce Figs 6, 7 and 8.** bestfits.csv contains the Table 1 data and both $\chi^2$ minimization. chi2s.py implementation of both $\chi^2$ functions described in the text. fig6.py code to reproduce Fig 6. fig7.py code to reproduce Fig 7. fig8.py code to reproduce Fig 8.
(ZIP)

## Author Contributions

**Data curation:** Johan Pallud.

**Formal analysis:** Stéphane Plaszczynski, Basile Grammaticos, Mathilde Badoual.

**Methodology:** Stéphane Plaszczynski, Basile Grammaticos, Jean-Eric Campagne, Mathilde Badoual.

**Software:** Stéphane Plaszczynski, Jean-Eric Campagne.

**Writing – original draft:** Stéphane Plaszczynski, Jean-Eric Campagne, Mathilde Badoual.

**Writing – review & editing:** Stéphane Plaszczynski, Basile Grammaticos, Johan Pallud, Jean-Eric Campagne, Mathilde Badoual.

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
