## [Decision Letter · Decision Letter 0]

3 Oct 2022

Dear Dr plaszczynski,

Thank you very much for submitting your manuscript "Predicting regrowth of low-grade gliomas after radiotherapy" for consideration at PLOS Computational Biology. As with all papers reviewed by the journal, your manuscript was reviewed by members of the editorial board and by several independent reviewers. The reviewers appreciated the attention to an important topic. Based on the reviews, we are likely to accept this manuscript for publication, providing that you modify the manuscript according to the review recommendations.

Sincerely,

Philip K Maini

Academic Editor

PLOS Computational Biology

Douglas Lauffenburger

Section Editor

PLOS Computational Biology

Reviewer's Responses to Questions

**Comments to the Authors:**

Reviewer #1: This paper deals with a very interesting topic in which the authors have made relevant contributions in the last decade. The paper is well written and the results very interesting since the authors confront a very simple model with data in order to anticipate regrowth after radiation therapy treatment for low-grade gliomas. I would have no objections for its publication in a different type of journal but not in PLOS computational Biology. The analysis in this manuscript is focused on the fitting of the patients data to Eq. (3) (or equivalently (4)), or the even simpler model given by Eq. (10). All of these equations are algebraic equations thus the computational part of this paper falls short of what one would expect for PLOS Computational Biology, no matter what is the interest of the results. Thus I recommend the authors to look for a different type of journal to report their results (e.g. a multidisciplinary one or a medical one emphasizing in the discussion the practical applicability of the results).

Reviewer #2: PCOMPBIOL-D-22-00915 Predicting regrowth of low-grade gliomas after radiotherapy

In this manuscript, Plaszczynski et al. present a mathematical model of low grade glioma growth and response to radiation therapy. They build on several years of work by their group and others to predict response to therapy and investigate the amount of data required to obtain a prediction on a patient-specific basis. The authors applied the model to data collected from 20 patients treated with radiation therapy at a single center in France. The manuscript is well written and the figures are generally well presented. The work will likely be of interest to the more mathematically inclined readership of Plos Computational Biology. I find no technical problems with the work. The following minor comments are intended to improve the clarity and presentation of the work.

>Section 1.4 appears to be incorrectly formatted. Perhaps this is meant to be a single paragraph?

>Pg 6 line 135 “It has no sense to use it on…” is awkwardly phrased

>Pg 7 line 158 “...quite precise, they lie within a few months” a prediction accurate ‘within a few months’ is arguably not too precise.

>Figure 6 caption: “This is repeated 1,000 times”.... Can the authors provide a specific rationale for this number of monte carlo simulations? Recognizing that this is simply a choice, was the limitation computational time? A sufficient characterization of the distribution? Randomly selected?

>Figures would benefit from more clear axis labels and legends embedded within the image. For example, axis labels and units for figure 2, meaning of colored lines in figures 6,8

>Data, but not computational codes are provided

**Have the authors made all data and (if applicable) computational code underlying the findings in their manuscript fully available?**

Reviewer #1: Yes

Reviewer #2: **No: **data is provided but not computational codes

PLOS authors have the option to publish the peer review history of their article (what does this mean?). If published, this will include your full peer review and any attached files.

Reviewer #1: No

Reviewer #2: No

Figure Files:

Data Requirements:

Reproducibility:

References:

---

## [Decision Letter · Decision Letter 1]

7 Nov 2022

Dear Dr plaszczynski,

Thank you very much for submitting your manuscript "Predicting regrowth of low-grade gliomas after radiotherapy" for consideration at PLOS Computational Biology. As with all papers reviewed by the journal, your manuscript was reviewed by members of the editorial board and by several independent reviewers. The reviewers appreciated the attention to an important topic. Based on the reviews, we are likely to accept this manuscript for publication, providing that you modify the manuscript according to the review recommendations.

Sincerely,

Philip K Maini

Academic Editor

PLOS Computational Biology

Douglas Lauffenburger

Section Editor

PLOS Computational Biology

Reviewer's Responses to Questions

**Comments to the Authors:**

Reviewer #1: This manuscript deals with an interesting and relevant problem: the prediction of regrowth of low-grade gliomas after radiotherapy. Although the methods used are simple computationally speaking, the result is interesting enough to warrant publication. I recommend it for publication although I think it may benefit from incorporation of several elements into the discussion:

1) Discussing to what extent radiation therapy alone is still used for treatment of low-grade gliomas and how the method could be extended to deal with combinations of radiotherapy and chemotherapies.

2) The weakest point of the manuscript in my opinion is the fact that the relationship between tmin and vd is purely empirical. I wonder if using a simplified tumor response to radiation therapy may help in obtaining an explanation for that law, e.g. something such as the calculation of the nadir in Lorenzo et al, J Roy Soc Interf (2019), doi:10.1098/rsif.2019.0195 that describes also response to radiotherapy for a type of slowly-growing, although different, cancer. Indeed that formula is also a time to progression as well as the one derived for brain metastasis after radiation therapy in Leon-Triana et al, doi:10.3390/math9070716. It may be relevant to discuss the relationship of the approach contained in the manuscript with those explicit formulae.

Reviewer #2: All concerns addressed. Thank you.

**Have the authors made all data and (if applicable) computational code underlying the findings in their manuscript fully available?**

Reviewer #1: Yes

Reviewer #2: **No: **Codes for figures are shared but source data is not.

PLOS authors have the option to publish the peer review history of their article (what does this mean?). If published, this will include your full peer review and any attached files.

Reviewer #1: No

Reviewer #2: No

Figure Files:

Data Requirements:

Reproducibility:

References:

---

## [Decision Letter · Decision Letter 2]

4 Mar 2023

Dear Dr plaszczynski,

We are pleased to inform you that your manuscript 'Predicting regrowth of low-grade gliomas after radiotherapy' has been provisionally accepted for publication in PLOS Computational Biology.

Best regards,

Philip K Maini

Academic Editor

PLOS Computational Biology

Douglas Lauffenburger

Section Editor

PLOS Computational Biology

Reviewer's Responses to Questions

**Comments to the Authors:**

Reviewer #1: The authors have properly considered all comments. I think the paper is now ready for publication.

**Have the authors made all data and (if applicable) computational code underlying the findings in their manuscript fully available?**

Reviewer #1: Yes

PLOS authors have the option to publish the peer review history of their article (what does this mean?). If published, this will include your full peer review and any attached files.

Reviewer #1: No

---

## [Editor Report · Acceptance letter]

28 Mar 2023

PCOMPBIOL-D-22-00915R2 

Predicting regrowth of low-grade gliomas after radiotherapy

Dear Dr Plaszczynski,

I am pleased to inform you that your manuscript has been formally accepted for publication in PLOS Computational Biology. Your manuscript is now with our production department and you will be notified of the publication date in due course.

With kind regards,

Anita Estes
